# Oral and Periodontal Implications of Hepatitis Type B and D. Current State of Knowledge and Future Perspectives

**DOI:** 10.3390/jpm12101580

**Published:** 2022-09-26

**Authors:** Dorin Nicolae Gheorghe, Francesco Bennardo, Dora Maria Popescu, Flavia Mirela Nicolae, Claudiu Marinel Ionele, Mihail Virgil Boldeanu, Adrian Camen, Ion Rogoveanu, Petra Surlin

**Affiliations:** 1Department of Periodontology, Faculty of Dental Medicine, University of Medicine and Pharmacy of Craiova, 200349 Craiova, Romania; 2School of Dentistry, Department of Health Sciences, Magna Graecia University of Catanzaro, Viale Europa, 88100 Catanzaro, Italy; 3Department of Gastroenterology, Faculty of Medicine, University of Medicine and Pharmacy of Craiova, 200349 Craiova, Romania; 4Department of Immunology, Faculty of Medicine, University of Medicine and Pharmacy of Craiova, 200349 Craiova, Romania; 5Department of Oral Surgery, Faculty of Dental Medicine, University of Medicine and Pharmacy of Craiova, 200349 Craiova, Romania

**Keywords:** periodontitis, hepatitis B, hepatitis D, pathogenesis, relationship, link, influence

## Abstract

Periodontitis is characterized by low-grade inflammation of the periodontal tissues, the structures that support and connect the teeth to the maxilla and mandible. This inflammation is caused by the accumulation of subgingival bacterial biofilm and gradually leads to the extensive damage of these tissues and the consequent loss of teeth. Hepatitis B is a major global health concern; infection with the hepatitis B virus causes significant inflammation of the liver and the possibility of its gradual evolution to cirrhosis. Hepatitis D, caused by infection with the delta hepatitis virus, is manifest only in patients already infected with the type B virus in a simultaneous (co-infected) or superimposed (superinfected) manner. The dental and periodontal status of patients with hepatitis B/D could exhibit significant changes, increasing the risk of periodontitis onset. Moreover, the progression of liver changes in these patients could be linked to periodontitis; therefore, motivating good oral and periodontal health could result in the prevention and limitation of pathological effects. Given that both types of diseases have a significant inflammatory component, common pro-inflammatory mediators could drive and augment the local inflammation at both a periodontal and hepatic level. This suggests that integrated management of these patients should be proposed, as therapeutical means could deliver an improvement to both periodontal and hepatic statuses. The aim of this review is to gather existing information on the proposed subject and to organize significant data in order to improve scientific accuracy and comprehension on this topic while generating future perspectives for research.

## 1. Introduction

The oral cavity hosts over 700 bacterial species, which usually co-exist in a harmonious state, called eubiosis when commensal bacteria do not allow harmful ones to trigger diseases [1]. These bacteria can also be found inside the gingival groove, or sulcus, a narrow space delimited by the tooth’s surface and the gingiva [2]. If the gingival sulcus is not properly and periodically cleaned by professional and at-home methods, this will allow the emergence of highly pathogenic bacteria. Consequently, this subgingival pathogenic bacterial biofilm will cause periodontal inflammation (periodontitis) [3]. In other words, if the subgingival biofilm is left undisturbed for lengthy periods of time, allowing highly pathogenic bacteria to colonize, the conditions for the onset of periodontitis are met [4]. As a result, these bacteria and their toxins reach the gingival tissues, causing the inflammatory response that is characteristic of periodontitis [3]. This low-grade, local inflammation usually has a gradual evolution, generating a damaging setting due to acidosis and enzyme activation for crucial elements of the periodontium, such as collagen fibers [5]. These fibers are the main component of the periodontal ligament, the structure of which connects the tooth to the alveolar bone. If damaged, the ligament will contribute to the formation of periodontal pockets (deep areas along the tooth’s root), which provide the ideal environment for more pathogenic bacteria [5]. Eventually, the alveolar bone itself is targeted by these bacteria and, under the effect of their collagenolytic enzymes and cellular damage, will begin to lose its normal size [6]. Consequently, the teeth will lose their support, increase their mobility, and finally be extracted, with significant consequences on the patient’s life quality and general health [6]. In addition, it is important to highlight that it is not only natural teeth that can be affected by the inflammation of their supporting tissues but dental implants too. In this case, the disease is called “peri-implantitis” and can lead to increased implant mobility, poor bone integration and, in some cases, implant removal [7].

Extensive research performed during the last few decades has shown that the consequences of periodontitis go beyond the disruption of normal dental functions [8]. The periodontium is linked to the rest of the body by blood and lymphatic means [9]. As a result, every pathologic alteration in general homeostasis has the potential to affect periodontal health [5]. Periodontitis, on the other hand, can affect a patient’s overall health, as well as the clinical presentation of specific conditions [5]. Researchers have investigated the bi-directional relationship between periodontitis and systemic health and sickness, leading to the formation of the “periodontal medicine” concept [10]. This concept incorporates and discusses the mutually influencing interactions that occur between periodontitis and systemic illnesses such as diabetes and cardiovascular disease [11,12]. Other significant correlations have been highlighted between periodontitis and autoimmune diseases such as rheumatoid arthritis and psoriasis [13,14]. In 2018, with the new classification of periodontal diseases, this concept gained clinical relevance, as certain systemic conditions were found to significantly modify the severity and rate of progression of periodontal diagnosis [15].

Hepatitis B is an infectious illness that damages the liver, caused by the hepatitis B virus (HBV) [16]. The virus is spread by contact with infected blood or bodily fluids [17]. In locations where the illness is widespread, infection around the time of birth or when in contact with other people’s blood throughout infancy are the most typical ways of contracting hepatitis B [17]. In locations where the illness is uncommon, the most common sources of transmission are intravenous drug use and sexual contact. Working in healthcare, blood transfusions, dialysis, living with an infected person, and traveling to countries with high infection rates are also considered to be significant risk factors [18]. HBV is capable of causing both acute and chronic infection [19]. Many people have no symptoms when they first become infected [20]. During an acute infection, some people may experience vomiting, yellowish skin, weariness, black urine, and abdominal discomfort [18,20]. These symptoms usually last a few weeks, and the first infection is seldom fatal [20]. Once infected, symptoms may develop from 30 to 180 days later [20,21]. If entering a chronic phase, the infection can lead to life-threatening complications, including cirrhosis or hepatocellular carcinoma [22].

Hepatitis D is caused by infection with the hepatitis D virus (HDV) and only occurs in individuals who are already infected with the HBV type [23]. HDV transmission can occur either concurrently with HBV infection (co-infection) or is superimposed on chronic hepatitis B or hepatitis B carrier status (superinfection) [23]. Because of the severity of its effects, an HDV infection in a person with chronic hepatitis B (superinfection) is considered the most dangerous kind of viral hepatitis [24]. In acute infections, these problems include an increased chance of liver failure and a rapid development of liver cirrhosis, as well as an increased risk of developing liver cancer in chronic infections [25]. Hepatitis D has the greatest mortality rate of any hepatitis infection, at 20% when combined with the hepatitis B virus [26]. According to a 2020 prediction, 48 million people are now afflicted with this virus [27].

Previous research was performed on the analysis of possible pathogenic connections existing between periodontitis and chronic hepatitis C (CHC) caused by the infection with the hepatitis C virus (HCV) [28,29]. Thus, it was highlighted that patients with CHC could exhibit significantly more severe oral health challenges, including periodontal ones, caused by the clinical manifestations of periodontitis (gingival bleeding, pocket depth, attachment loss) when compared to healthy controls. Local periodontal inflammation has been shown to have increased strength in CHC patients, as depicted by the immunological quantitative assessment of relevant pro-inflammatory mediators in gingival crevicular samples (GCF) [29]. These mediators include interleukins (IL, such as IL-1α, IL-1β, IL-18), inflammasomes (NLRP3 inflammasome), collagenolytic enzymes (Caspase-1) and pentraxins (PTX, such as PTX-3 and C-reactive protein). All of these pro-inflammatory markers were shown to express more elevated GCF levels in patients with periodontitis and CHC than in periodontitis patients with no CHC or with healthy controls [28]. Interestingly, these markers have also been shown to express serum-elevated levels in CHC patients, suggesting an additional periodontal risk in such patients [29]. Conversely, the implementation of non-surgical periodontal therapy in these patients has delivered less significant improvements in the intensity of local periodontal inflammatory reaction than in non-CHC periodontitis patients, suggesting a limited efficiency of the therapy in their case. Thus, an additional therapeutical focus should be given to these types of patients when seeking periodontal or dental care [28,29].

Given the results generated by our previous research on the topic of possible pathogenic connections existing between HCV infection and periodontitis, we aim to expand the project on HBV and HDV infection and periodontal links. Thus, we performed a review of the existing relevant scientific literature in order to gather data and assess the current state of the art. The aim of this review is to extract and compile available information on the subject so as to set future development of this topic and to generate the scientific background needed for the onset of future projects on possible pathogenic connections existing between periodontitis and HBV/HDV infection.

## 2. Materials and Methods

This review followed the criteria and guidelines of the Preferred Reporting Items for Systematic Review and Meta-Analyses (PRISMA) (Figure 1).

### 2.1. PICO Question

“Is there currently relevant scientific data on the possible significant pathogenic connections between oral implications and HBV/HDV infection that could offer the background for future development of the subject?” (Population: patients with hepatitis B/D infection; Intervention: oral health status assessment; Comparison: relevant background information; Outcome: development of complementary studies).

### 2.2. Search Strategy

Relevant scientific databases were electronically searched for English in-extenso papers and abstracts for this review. These included Medline (via PubMed), Web of Science, and Scopus. The time bracket for the search was set between 1st January 1970 and 1st July 2022. The keywords used during the search were: “periodontitis”, “hepatitis B”, and “hepatitis D”, “oral”, “oral health”, “oral status”, “periodontal disease”, “hepatitis B virus”, “hepatitis D virus”, “pathogenic”, “connection”, “patients”, “gingival fluid”, “dental status”, “transmission”, “saliva” and “blood”, using the Boolean operators “AND”, “OR”.

### 2.3. Exclusion Criteria of the Studies

Some of the generated search articles were excluded on the following premises: (1) they reported in vitro and experimental animal studies, (2) self-reported studies focusing on dental practitioners and dental students’ knowledge of the transmission of the virus, and (3) letters and editorials. All other remnant studies were included for further analysis.

### 2.4. Information Extraction and Review Structuring

The selected papers were carefully read, and relevant information for the review was extracted. The review consisted of three parts: (1) information on HBV infection and oral implications, (2) information on HDV infection and oral implications, and (3) future perspectives and proposed development of the subject.

## 3. Results

### 3.1. HBV Infection and Oral Implications

The search for papers on HBV infection and oral implications returned 58 results, ranging from 1977 to 2022. After applying the exclusion criteria, 16 papers were finally selected for critical reading and idea synthesis. Additional papers were extracted from these papers’ references list, if they were not generated by the database search.

The earliest papers on the subject were published between 1984–1985, focusing on the detection of HBV antigens (or viral particles) in the oral fluids of infected patients. The study by Polloch et al. identified HBV surface antigens (HBsAg) in the GCF samples of HBV-infected patients [30]. In 1985, Ben-Aryeh et al. performed a similar study, which concluded that HBsAg was found in 90% of the assessed GCF samples originating from HBV-infected patients [31]. The surface antigen was also found in the saliva samples of the same patients. It could be speculated that the saliva might have been contaminated with the blood of gingival origin due to gingival inflammation. However, the authors found no significant correlation between the presence of HBsAg in the saliva samples and the gingival status of the participating patients in terms of gingival inflammation or the gingival bleeding index [31]. This led the authors to suggest that the source of HBsAg presence in the saliva samples was, in fact, the GCF [28]. The virus circulates in the general bloodstream, reaches the lymphatic system and eventually into the GCF due to a difference in osmotic pressure. Then it reaches the saliva, using the GCF as a carrier [32]. This hypothesis is also endorsed by the 1984 study by Hurlem et al., who suggested that HBV-infected patients may pose a higher risk of viral transmission in the dental office by double carriers: saliva and increased gingival bleeding when dealing with gingival inflammation [33].

This hypothesis is further emphasized by a more recent study by Kamimura et al., who found a strong correlation between occult blood traces in saliva and the presence of viral HBV particles [34]. This study highlighted how HBV DNA particles were found in saliva samples, particularly of elderly patients diagnosed with periodontitis [34]. The study further speculates that this may pose an increased risk of horizontal HBV transmission in the family, where the probability of contact with infected saliva is quite elevated [34]. The risk is further enhanced if patients suffer from periodontitis, with an increased gingival bleeding index [34]. A study by Farghaly reached similar conclusions suggesting that patients with periodontitis showed a higher proportion of hepatitis exposure and a higher detectability of salivary HBsAg [35]. Additional risk factors were considered to be the rural residence of patients or a medical history of past blood transfusions. Thus, the study concluded that the presence of periodontitis, severe gingival bleeding and poor oral hygiene were associated with the risk of hepatitis and the detectability of salivary hepatitis markers [35]. Similar results were generated in the study by Sharifian et al., who considered that the most frequent risk factors for HBV infection in studied patients were positive periodontal diagnosis and family history [36].

The clinical settings of unfavorable dental and periodontal diagnosis and liver damage were assessed by Yang et al., who concluded that an increased number of absent teeth were associated with an increased risk of primary liver cancer [37]. A study by Nagao et al. also highlighted that periodontitis might be correlated with viral liver disease [38]. However, the results seem inconclusive so far, as a 2011 study by Anand et al. found that the number of dental caries and the periodontal status of patients with nonalcoholic cirrhosis did not differ significantly from that of the controls without any liver disease [39]. Nevertheless, other oral health issues, such as halitosis, have been directly linked to HBV infection and periodontitis, including a study by Hun Han et al. [40]. The authors concluded that patients with periodontitis, HBV infection and neglected tongue-brushing had the highest prevalence of volatile sulphur halitosis, suggesting that liver function should be evaluated in patients dealing with bad breath [40].

The periodontal management of patients with an HBV infection was studied by Seshima et al., who reported a case of effective, regenerative periodontal therapy [41]. The patient suffered from HBV infection and diabetes mellitus, which can significantly impact the body’s healing and regenerative capabilities [41]. However, considering the medical history of the patient, the authors reported a clinical improvement in the periodontal parameters [41]. A study by Ting et al. suggested the use of statins as an adjunctive to periodontal therapy in patients with an HBV infection [42]. This is justified by the antiviral properties of statins, as well as their antibacterial capabilities, including on important periodontal pathogens such as *Porphyromonas gingivalis* [42]. Concerning the surgical management of periodontal patients with an HBV infection, Hong et al. reported no episodes of postoperative bleeding in patients, despite a significant correlation of the international normalized ration (INR) with HBV infection diagnosis [43]. The authors suggested that it was not only INR values that should be considered when evaluating patients with liver diseases for procedures with a post-surgical bleeding risk [43].

From an immunological perspective, certain inflammation mediators were targeted in the saliva samples of patients with an HBV infection [44]. Pro-inflammatory interleukins (IL-2 and IL-4), as well as anti-inflammatory ones (IL-10), expressed significantly more elevated levels in the saliva samples of HBV infected patients than in the healthy controls, as depicted by the enzyme-linked immunosorbent assay (ELISA) used in this study [44]. The same immunological method (ELISA) was proposed in the study by Gharavi et al. as a diagnosis tool for HBV infection in samples of saliva, with good sensitivity and specificity [45].

### 3.2. HDV Infection and Oral Implications

The search for papers on the oral implications of HDV infections retrieved 13 papers, of which only 3 could be selected, for critical reading after applying the exclusion criteria. This low number of papers on the subject suggests a limited current understanding of the subject and should stimulate future developments of the topic.

In a 1986 article, Cottone et al. raised awareness among dental practitioners and members of the dental office team of the possibility of the transmission of the newly identified, at that time, HDV virus [46]. The authors stated that the hepatitis D virus could pose a serious threat to all members of the dental team and thus encouraged vaccination against the HBV virus, as it would also offer protection against HDV [46].

One of the main reasons why the literature on the oral implications of HDV is limited, is that the viral infection is mainly conditioned by a co-existing or pre-existing infection with HBV. Hence, the patient target group is limited only to HBV-positive persons. Even though the association of the HBV and HDV viruses is generally accepted and agreed upon, some authors have reported exceptions to this. In 2016, Weller et al. detected HDV in the salivary glands of Sjogren syndrome patients [47]. Their micro-array analysis showed that HDV was present in more than 50% of the samples originating from primary Sjogren syndrome patients [47]. The novelty of the study was the fact that the identification of HDV was independent of any HBV presence. This suggests that HDV is able to set up an independent presence without HBV, at least at the salivary gland level, and exhibits a unique tissue tropism [47]. The results of this study raise significant awareness from an oral health perspective, as Sjogren syndrome is considered to be a major trigger for dental and periodontal problems, as well as an extra-hepatic manifestation of liver diseases, including viral infections.

Currently, there is insufficient data on whether HDV particles could be carried by saliva, similar to HBV. Only one study, performed by Isaeva et al., focused on this topic but found no detection of HDV antibodies in saliva samples originating from patients with HBV and HDV infection [48]. Despite the fact that the saliva samples were positive for HBV antigens and antibodies, this was not the case for HDV, suggesting a lower concentration of these elements in the saliva than for HBV [48]. Nevertheless, the matter should be addressed by complementary research in order to increase its scientific understanding.

## 4. Future Perspectives

As shown by the literature review (Table 1), currently, there is sufficient data on the oral implications of HBV infection and little, or almost no insight, into these implications in HDV ones. HBV infection and oral implications cover mostly the detection of viral antigens in saliva and gingival fluid and less about the clinical, dental, or periodontal status of infected patients [49]. There is also a gap in the literature regarding the assessment of various pro-inflammatory elements in samples of gingival fluid or saliva, as this can have relevance for the characterization of low-grade inflammatory periodontal reactions and their elements in this type of patient. Regarding HDV infection, this has received little attention from the perspective of oral health implications, mainly because patients with HBV and HDV co-infection or supra-infection may be more difficult to gather for larger studies. The epidemiology of the HDV infection may vary significantly from region to region, and as HBV vaccinations continue to gain popularity, the spread of the HDV virus may also decelerate.

Considering the setting, pilot studies on smaller groups of patients could be generated in order to probe the particularities of oral and periodontal health in patients with an HBV + HDV infection. The first step of this project would be to compare clinical data on the oral and periodontal health status of HBV + HDV infected individuals, such as the number of missing teeth, periodontal diagnosis, and the type of diagnosed periodontal conditions, in terms of the severity and rate of progression, as compared to the controls. The ideal circumstance would be to include patients who do not suffer from other systemic diseases that could influence the manifestation of periodontitis (such as diabetes mellitus), but this would remain to be established by the study design and group characteristics [50]. An immunological analysis via the ELISA method would be necessary in order to measure specifically targeted pro-inflammatory mediators in GCF samples that have relevance in both periodontitis and HBV + HDV infections pathogenesis. The local and systemic effects of periodontal therapy in patients diagnosed with periodontitis and HBV + HDV infection should also be evaluated, from a clinical and immunological standpoint, in order to detect improvements in the expression of inflammatory mediators as a sign of the inflammatory reaction’s modulation.

As the prevalence of HDV infection in Romania experienced recent rising trends [51] and considering the general increase in population mobility after the COVID-19 pandemic, the development of such a research project could be of significant interest and deliver valuable and high-novelty results. With the experience gained from the previous HCV infection study, we plan to apply the same principles and management of the project in this new research direction in order to improve existing knowledge and increase scientific awareness of the topic.

## 5. Conclusions

The existing literature offers sufficient background information on the oral implications of HBV infection in order to fundamentally support the development of a research project on the topic of HBV + HDV co-infection, where data is scarce and has significant gaps.

## Figures and Tables

**Figure 1 jpm-12-01580-f001:**
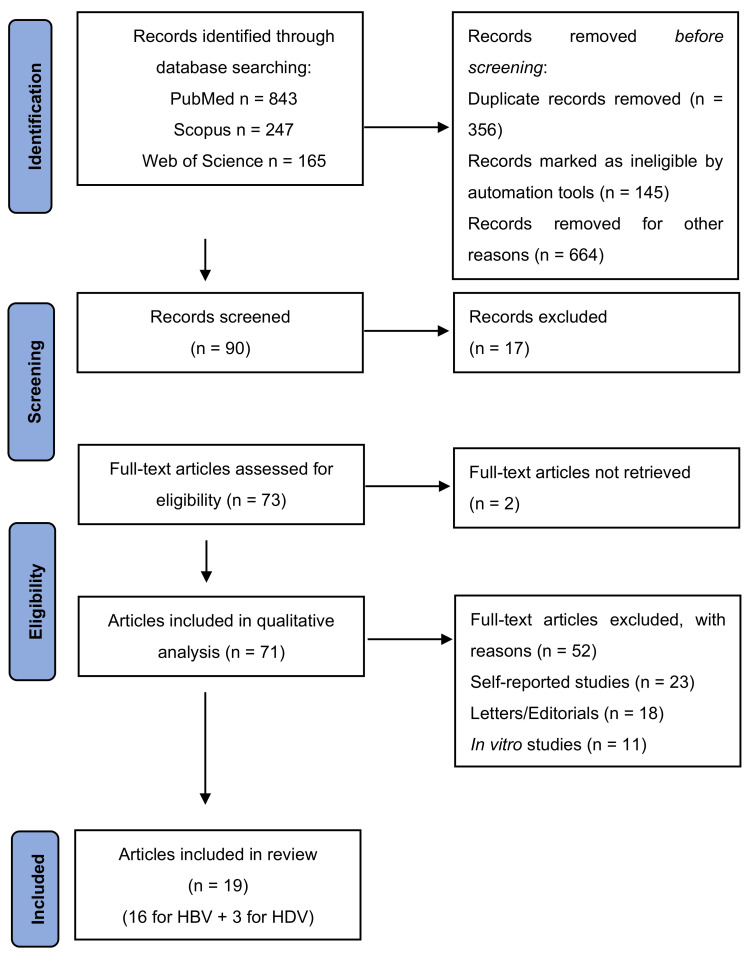
PRISMA flow diagram. HBV = hepatitis B virus; HDV = hepatitis D virus.

**Table 1 jpm-12-01580-t001:** Synopsis of the selected papers and their findings.

Reference	Hepatitis Virus Type	Findings
Polloch et al., 1984 [30]	HBV	HBsAg detection in gingival crevicular samples of infected patients
Hurlem et al., 1984 [33]	HBV	HBV infected patients may pose a higher risk of viral transmission in the dental office, via saliva and gingival bleeding
Ben-Aryeh et al., 1985 [31]	HBV	HBsAg detection in 90% of gingival crevicular fluid and saliva samples of infected patients
Cottone et al., 1986 [46]	HDV	HDV infection is a significant risk in the dental office, recommended HBV vaccination for members of the dental team
Farghaly et al., 1998 [35]	HBV	Patients with periodontitis showed a higher percentage of hepatitis exposure and a higher detectability of salivary HBsAg
Anand et al., 2001 [39]	HBV/HCV	Periodontal status of patients with nonalcoholic cirrhosis did not differ significantly from that of controls with no liver disease
Azatyan et al., 2001 [44]	HBV	Elevated expression of pro-inflammatory interleukins in saliva samples of HBV infected patients
Lamster et al., 2007 [32]	HBV/HCV	Gingival crevicular fluid could be an important source for traces of hepatitis viruses’ presence in saliva
Hong et al., 2012 [43]	HBV	No episodes of postoperative bleeding in periodontal management of HBV infected patients, independent of INR values
Hun Han et al., 2013 [40]	HBV	Direct link between volatile sulphur halitosis, periodontitis, and HBV infection
Nagao et al., 2014 [38]	HBV/HCV	Periodontitis might be correlated with viral liver disease
Ting et al., 2015 [42]	HBV	Use of statins as an adjunctive periodontal therapy in patients with HBV infection
Seshima et al., 2016 [41]	HBV	Effective and stable results of regenerative periodontal therapy in patients with HBV infection
Weller et al., 2016 [47]	HDV	Identification of independent HDV in 50% of salivary gland samples from HDV infected patients with primary Sjogren syndrome
Yang et al., 2017 [37]	HBV	Increased number of absent teeth is associated with increased risk of primary liver cancer
Sharifian et al., 2019 [36]	HBV	Risk factors for HBV infection: positive periodontal and family history
Gharavi et al., 2020 [45]	HBV	HBV can be detected in saliva samples of infected patients by means of ELISA method
Isaeva et al., 2020 [48]	HDV	No detection of HDV antibodies in saliva samples from patients with HBV and HDV infection
Kamimura et al., 2021 [34]	HBV	Strong correlation between occult blood traces in saliva and HBV presence in saliva samples

HBV—hepatitis B virus; HCV—hepatitis C virus; HDV—hepatitis D virus; HBsAg—hepatitis B surface antigen; ELISA—enzyme linked immunosorbent assay.

## Data Availability

Not applicable.

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
