# Peer review of "Oral and Periodontal Implications of Hepatitis Type B and D. Current State of Knowledge and Future Perspectives"

_jpm, 2022, doi:10.3390/jpm12101580_

Round 1

Reviewer 1 Report

Dear Authors, 

you made a great work!

However, some improvements can improve the manuscript.

Author Response

Dear Reviewer, thank you very much for your dedicated time and valuable suggestions in order to improve the manuscript. As many as possible of the comments have been applied to the revised version of the paper. Please find below the response to all the comments and suggestions. Kind regards!
The paper is a review on the oral and periodontal implications of Hepatitis Type B and D.
The Authors made a great work in terms of methodology and the paper sounds scientific and well
written. Thank you!
However, some improvements are mandatory before acceptance.
The abstract is well written, complete and summary in its various aspects. The keywords are
complete and appropriate. Thank you!
In the introduction:
• “If the gingival sulcus is not properly and periodically cleaned by professional and at-home
methods, this will allow the emergence of highly pathogenic bacteria. Consequently, this
subgingival pathogenic bacterial biofilm will cause periodontal inflammation (periodontitis)
[3].” not only this, but I believe it is important to underline how one of the most frequent
types of rehabilitation currently used in dentistry, maintains a more inflammatory
environment around it even in a condition of clinical health, and in any case always more
inflamed than neighboring teeth! I believe it is extremely important to consider this aspect,
as indicated by: doi: 10.3390/jpm12010058.” We thank the reviewer for highlighting this oversight of periodontal pathology. Information has been added, using suggested reference (Lines 60-64)
• the description of the pathology is well conducted and absolutely relevant. Thank you!
In materials and methods:
• “PICO” please explain in a more discursive way each letter of this acronym depends on the
question. Explanations for Population/Intervention/Comparison/Outcome has been added (Lines 140-142)
In the Results:
• please check Figure 1 quality. Done, new figure provided.
• they are well written and relevant. I suggest the authors to make this section more readable
to insert a schematic classification of the results. Done, synoptic table with findings added.
“Future perspectives” is really important section for this type of manuscript. I really appreciate. Thank you very much.
Conclusions are concise and clear. Thank you very much.
Bibliography is formatted respecting the journal’s requirements and no improper citations are
evidenced. Thank you.
Figures and labels are clear and easy to comprehend. Thank you.
English is clear and easy to understand. Thank you.

Reviewer 2 Report

Dear Authors!

Congratulations for the nice work and the well-written paper! I can agree with the main flow, however, please let me raise a few issues:

In the section, starting in line 74, I miss a few sentences and references about periodontitis and general medical disease, such as RA, psoriasis, diabetes, cardiovascular disease. Please add at least some of the following references:

Antal M, Battancs E, Bocskai M, Braunitzer G, Kovács L. An observation on the severity of periodontal disease in past cigarette smokers suffering from rheumatoid arthritis- evidence for a long-term effect of cigarette smoke exposure? BMC Oral Health. 2018 May 10;18(1):82. doi: 10.1186/s12903-018-0531-5. PMID: 29747598; PMCID: PMC5946453.

Gheorghita D, Antal MA, Nagy K, Kertesz A, Braunitzer G. Smoking and Psoriasis as Synergistic Risk Factors in Periodontal disease. Fogorv Sz. 2016 Dec;109(4):119-124. English, Hungarian. PMID: 29949256.

Antal M, Braunitzer G, Mattheos N, Gyulai R, Nagy K. Smoking as a permissive factor of periodontal disease in psoriasis. PLoS One. 2014 Mar 20;9(3):e92333. doi: 10.1371/journal.pone.0092333. Erratum in: PLoS One. 2014;9(10):e110975. PMID: 24651659; PMCID: PMC3961310.

Gheorghita D, Eördegh G, Nagy F, Antal M. A fogágybetegség mint az atheroscleroticus cardiovascularis betegség rizikófaktora [Periodontal disease, a risk factor for atherosclerotic cardiovascular disease]. Orv Hetil. 2019 Mar;160(11):419-425. Hungarian. doi: 10.1556/650.2019.31301. PMID: 30852909.

Battancs E, Gheorghita D, Nyiraty S, Lengyel C, Eördegh G, Baráth Z, Várkonyi T, Antal M. Periodontal Disease in Diabetes Mellitus: A Case-Control Study in Smokers and Non-Smokers. Diabetes Ther. 2020 Nov;11(11):2715-2728. doi: 10.1007/s13300-020-00933-8. Epub 2020 Sep 25. PMID: 32975709; PMCID: PMC7547922.

On Figure 1. PRISMA flow diagram.  There are some problems: In the upper right box there is another box, and this way it clashes with the box below. Please correct it.

It is somewhat disturbing, that on Figure 1 the results of the PRISMA  flow are shown, but in the text the result of HDV and HBV are explained separately. It would be better to have an image for this, or include these data also for the image…

Author Response

Dear Reviewer, thank you very much for your dedicated time and valuable suggestions in order to improve the manuscript. As many as possible of the comments have been applied to the revised version of the paper. Please find below the response to all the comments and suggestions. Kind regards! Dear Authors! Congratulations for the nice work and the well-written paper! I can agree with the main flow, however, please let me raise a few issues: Thank you very much! In the section, starting in line 74, I miss a few sentences and references about periodontitis and general medical disease, such as RA, psoriasis, diabetes, cardiovascular disease. Please add at least some of the following references: We thank the reviewer very much for highlighting this oversight. Information has been added, supported by the suggested references (Lines 74-76) On Figure 1. PRISMA flow diagram. There are some problems: In the upper right box there is another box, and this way it clashes with the box below. Please correct it. Done, corrections applied to Figure 1. It is somewhat disturbing, that on Figure 1 the results of the PRISMA flow are shown, but in the text the result of HDV and HBV are explained separately. It would be better to have an image for this, or include these data also for the image…Thank you reviewer for pointing out this aspect. We chose this solution for illustration of the results because the search of papers on HDV infection and oral health problems retrieved very little items. Thus, we considered that a separate PRISM diagram just for HDV infection would have been redundant, hence choosing the variant of a merged HBV and HDV Figure.

Reviewer 3 Report

Well written, it is a qualified systematic review. The study followed the Preferred Reporting Items for Systematic Review and Meta-Analyses (PRISMA) criteria and guidelines.

Minor comments:

1. Please include the number of final HBV and HDV studies in Figure 1. PRISMA flow diagram. 

2. Is there any paper studying with other types of virus. If yes, please include in the review.

Author Response

Dear Reviewer, thank you very much for your dedicated time and valuable suggestions in order to improve the manuscript. As many as possible of the comments have been applied to the revised version of the paper. Please find below the response to all the comments and suggestions. Kind regards! Well written, it is a qualified systematic review. The study followed the Preferred Reporting Items for Systematic Review and Meta-Analyses (PRISMA) criteria and guidelines. Thank you! Minor comments: 1. Please include the number of final HBV and HDV studies in Figure 1. PRISMA flow diagram. Done, numbers added in “Included” box of Figure 1. 2. Is there any paper studying with other types of viruses. If yes, please include in the review. Thank you, reviewer, for this suggestion. The review includes relevant previous studies on the oral and periodontal implications of hepatitis C virus infection in the “Introduction” section. Since this review focused only on hepatitis viruses, we plan to develop a new review in the future, following your suggestion, on oral and periodontal implications of a variety of other viruses, such as HIV, Epstein Barr, Covid-19 etc. Thank you for this lucrative idea!